# Etiology and Multi-Drug Resistant Profile of Bacterial Infections in Severe Burn Patients, Romania 2018–2022

**DOI:** 10.3390/medicina59061143

**Published:** 2023-06-14

**Authors:** Bogdan Nițescu, Daniela Pițigoi, Daniela Tălăpan, Maria Nițescu, Sorin Ștefan Aramă, Bogdan Pavel, Adrian Streinu-Cercel, Alexandru Rafila, Victoria Aramă

**Affiliations:** 1Faculty of Medicine, “Carol Davila” University of Medicine and Pharmacy, 050474 Bucharest, Romania; 2Clinical Emergency Hospital of Plastic, Reconstructive Surgery and Burns, 010761 Bucharest, Romania; 3“Prof. Dr. Matei Balș” National Institute of Infectious Diseases, 021105 Bucharest, Romania; 4Faculty of Dentistry, “Carol Davila” University of Medicine and Pharmacy, 050474 Bucharest, Romania

**Keywords:** burns, bacterial infections, multi-drug resistance, *Pseudomonas aeruginosa*, wound infections

## Abstract

Infections in severe burns and their etiology are and will remain a big concern for the medical field. The multi-drug resistant strains of bacteria are a challenge of today’s medicine. The aim of our study was to identify the etiological spectrum of bacterial infections in severe burn patients in Romania and their multi-drug resistant patterns. We performed a prospective study that included 202 adult patients admitted to the intensive care unit (ICU) of the Clinical Emergency Hospital of Plastic, Reconstructive Surgery and Burns, Bucharest, Romania (CEHPRSB), from 1 October 2018 to 1 April 2022, a period which includes the first 2 years of the outbreak of COVID-19. From each patient, wound swabs, endotracheal aspirates, blood for blood culture, and urine were collected. The most frequently isolated bacterium was *Pseudomonas aeruginosa* (39%), followed by *Staphylococcus aureus* (12%), *Klebsiella* spp. (11%), and *Acinetobacter baumannii* (9%). More than 90% of *Pseudomonas aeruginosa* and *Acinetobacter baumannii* were MDR, regardless of the clinical specimen from which they were isolated.

## 1. Introduction

The burn patient represents one of the most complex problems for an intensive care unit medical doctor. Apart from electrolytic and hemodynamic resuscitation, nutrition support, and peri-surgical management, multi-drug resistance increases the difficulty of treatment and may lead to sepsis and multiple organ system failure, and it will severely impair the outcome of the prognosis.

Multi-drug resistant bacteria (MDR) and antimicrobial resistance (AMR) are included in the top 10 public health concerns facing humanity [1]. Drug resistance is life-threatening in patients with impaired immunological response pathologies, but associated with severe burns, and it becomes an even bigger problem due to loss of the skin barrier, which is our primary defense against bacteria. Severe burns are defined as Total Body Surface Area (TBSA) greater than 20%, regardless of the body region affected, and excluding first-degree burns—sunburns. Severe burns can also be defined as inhalation burns, regardless of the TBSA, and represent one of the most challenging pathologies. The progress made in the treatment of burns has been significant in the last three decades, but that is not enough considering that the majority of the severe burn patients admitted to intensive care units (ICU) must survive bacterial infection.

The most common associated bacteria and fungi are *Acinetobacter baumannii, Pseudomonas aeruginosa*, *Staphylococcus aureus* (methicillin-resistant *Staphylococcus aureus*—MRSA), and *Candida* spp. [2]. Until the 21st day of hospital rest, 57% of the open burn wounds will be colonized with *extended-spectrum β-lactamase Pseudomonas*. Additionally, 30% of patients with >40% TBSA will develop *Candida* colonization or infections. *Pseudomonas* and MRSA are most common in the United States, whereas in tropical regions, such as Singapore, due to humid environmental factors, *Acinetobacter baumannii* is the leading cause of infections [2,3,4,5,6,7]. In the United States, reports between 2001 and 2010 from the National Burn Repository (NBR) indicate that 90% of the burns had less than 20% TBSA from 163,771 patients, which means that severe burns represent a rare pathology. The mortality ranged from 0.6% in less than 10% TBSA to 74% in more than 80% TBSA. A 23% mortality rate was recorded in patients associated with inhalation injury. Small children, people with disabilities, and military personnel are more prone to burn injuries, which are also more likely in developing countries than in developed countries [2].

Complications in severe burns are in the majority of healthcare-associated infections (HAI). Invasive burn complications due to bacterial infections are most commonly in order of decreasing frequency: pneumonia, cellulitis, urinary tract infections, respiratory failure, wound infections, and bloodstream infections (in association with central catheterization). Other types of infections include ophthalmic infections (in burns associated with eye involvement), chondritis (in burns that affect the external ear), gastrointestinal tract infections (including *Clostridioides difficile*), and tetanus [2,5]. Infections in burn patients have different bacterial etiologies in relation to the length of stay in hospitals. Initially, Gram-positive organisms such as *Staphylococcus aureus* tend to colonize and infect the burn wound, whereas, after 3 weeks, the incidence of *Pseudomonas aeruginosa* increases in wound cultures. A study performed in a Canadian burn center on 125 patients between 2010 and 2013 demonstrated that *Pseudomonas aeruginosa* was found in only 8% of wound cultures in the first 7 days after hospital admission. The incidence increased to 55% in the fourth week of hospital stay, thus demonstrating that regardless of hygiene and epidemiological prevention, severe burn wounds are extremely susceptible to infections [8,9,10]. One of the most interesting wound infections that is very difficult to treat is the co-existence of a mixed biofilm of *Pseudomonas* and *Staphylococcus*. Even if one is Gram-positive and one is Gram-negative, there have been studies that indicate that not only they can co-exist in the same wound, but they enhance each other to infect the wound. They co-infect the human keratinocytes and inhibit wound closure, sustaining an inflammatory response and allowing bacterial colonization and infection [11].

The joint meeting between the Centers for Disease Control and Prevention (CDC) and European Centre for Disease Prevention and Control (ECDC) tried to set up a consensus definition for the terms related to drug resistance. The MDR was defined as bacteria resistant to three or more antimicrobial classes [12]. Today, CDC uses this term to refer to a microorganism that is resistant to at least one antibiotic in three or more drug classes [13].

Our attention was drawn to the treatment of burn infections, local or systemic, rather than the practice of prevention and surgical treatment. The latter are also important in reducing wound colonization, but even if the guidelines of hospital hygiene are thoroughly respected, infection of severe burn wounds in patients with prolonged hospital stays is, to some degree, unavoidable.

The main goal of this study was to identify the etiological spectrum of bacterial infections in severe burn patients in Romania and their multi-drug resistant patterns. Our hospital has the only ICU in the country dedicated entirely to burn care and has over 60% of patients annually transferred from other hospital units around the country. Starting with this 200+ patient study conducted over three and a half years, we can correct and implement healthcare strategies to improve the outcome of severe burn patients and increase our confidence by comparing our findings to similar data from highly developed countries. Unfortunately, the burn treatment lacks data in comparison to other pathologies, with very few studies being performed on burn infections in Eastern Europe in the last 20 years.

## 2. Materials and Methods

### 2.1. Study Design and Study Setting

This is a prospective study that included 202 adult patients admitted to the ICU of the Clinical Emergency Hospital of Plastic, Reconstructive Surgery and Burns, Bucharest, Romania (CEHPRSB), from 1 October 2018 to 1 April 2022, a period which includes the first 2 years of the outbreak of COVID-19. The CEHPRSB is the only plastic-surgery-dedicated hospital in Romania, having the only ICU at a national level exclusively dedicated to burn treatments (until 2020), treating only adult patients. The ICU unit has 5 hospital beds in 5 different hospital rooms. The hospital also has one burn ward for mild burn patients.

### 2.2. Study Population and Specimen Collection

The patients admitted to the ICU are categorized as severe burns, the criteria for admission being TBSA over 20% of minimum 2A degree burns and/or inhalation injury regardless of the TBSA.

The exclusion criteria in this study included patients with less than 4 days of hospital stay and patients who had severe immunodeficiency or severe associated pathologies (e.g., HIV infection, autoimmune diseases, coagulopathies).

The number of patients included in this study represents over 90% of the patients admitted in this 3.5-year period, and it is representative at a national scale because, as mentioned earlier, our hospital has the only exclusive burn-dedicated ICU, patients from all over the country being admitted in our unit. It must be mentioned also that this period included the COVID-19 pandemic period, which made the research difficult due to short periods of staff quarantine, which made the unit non-functional for 2 months in total.

Our surgical protocol is early excision and consecutive skin grafting as soon as the biological status of the patient is in parameters, and it is repeated as soon as possible until healing. The patient’s dressings are changed daily in a sterile environment, and a therapeutic bath is performed. Before and between surgical times, the wounds are dressed in silver sulfadiazine creams.

There is no prophylactic antibiotic treatment at admission, so the patients did not receive antibiotics before collecting the biological samples for the microbiology laboratory. We do use 1 dose of antibiotic at the time of surgery. Our hospital protocol includes a collection of skin swabs at admission to detect colonization (not the subject of our study, so were excluded). Since the literature shows that invasive burn complications due to bacterial infections are most frequently wound infections, pneumonia, urinary tract infections, and bloodstream infections [2,5], we focused our attention on the following four types of biological samples: burn wound swabs, endotracheal aspirates, urine, and blood for blood culture. Burn wound swabs were collected at admission and repeated throughout the whole admission twice a week (Monday and Thursday) [14]. If a patient had an inhalation injury, the endotracheal aspirates were also collected at admission and repeated throughout the whole admission twice a week (Monday and Thursday). Urine for urine culture was obtained once a week (Wednesday) according to our ICU protocol, and blood for blood culture was drawn only if the patients had a body temperature >38.5 °C (Figure 1). Clinical specimens were collected every day of the week between 7 a.m. and 9 p.m. (laboratory working hours) and sent immediately to the microbiology laboratory.

In this study, all microorganisms were isolated during the study period from the patients admitted to the CEHPRSB, choosing one different microorganism per sample and patient.

Patient data were obtained from the medical records of each patient—no further medical investigations were necessary except the normal ICU protocol.

### 2.3. Laboratory Methods

All specimens were processed immediately after they were received in the laboratory.

Blood culture bottles (VersaTREK REDOX 1 and REDOX 2) were incubated (for 5 days at 37 °C) and identified as positive by the Thermo Scientific VersaTREK^®^ automated microbial detection system (TREK Diagnostic Systems, West Sussex, England, UK). Once the specimens were positive, the microbiologist performed a Gram stain and placed them on agar plates: sheep blood agar and Drigalski agar (S.C. Medical Psiho Group S.R.L., Hunedoara, Romania). Plates were then incubated for 24 h at 37 °C, and cultures growing on them underwent pathogen identification by MicroScan^®^ Walkaway 40 Plus (Beckman-Coulter Inc., West Sacramento, CA, USA)—automated biochemical identification. Antimicrobial susceptibility testing (AST) was performed by using lyophilized minimum inhibitory concentration (MIC) micro-broth dilution plates NUC57 (for Gram-negatives) and POS32 (for Gram-positives) at the MicroScan^®^ Walkaway 40 Plus (Beckman-Coulter Inc., West Sacramento, CA, USA) automated device.

Burn wound swabs, endotracheal aspirates, and urine for urine culture were processed according to the CEHPRSB microbiology laboratory protocol for bacteria and following a Chinese guideline for fungi [15]. Clinical specimens were inoculated on ready-to-use non-selective and selective media and then were incubated at 37 °C for 18–24 h. There were used sheep blood agar (as non-selective), Drigalski agar (selective for Gram-negatives), mannitol salt agar (Chapman’s agar, selective for *Staphylococcus aureus* and coagulase-negative staphylococci) and Sabouraud agar (selective for fungi) (S.C. Medical Psiho Group S.R.L., Hunedoara, Romania). Isolates were identified and tested for antimicrobial susceptibility using the appropriate plates (NUC57 and POS32) with MicroScan^®^ Walkaway 40 Plus (Beckman-Coulter Inc., West Sacramento, CA, USA) system. Classes of antibiotics tested for Gram-negatives (NUC57) were the following: penicillins (piperacillin-tazobactam, ticarcillin-clavulanic acid), cephalosporins (ceftazidime, cefepime), carbapenems (imipenem), monobactams (aztreonam), fluoroquinolones (ciprofloxacin), aminoglycosides (amikacin, gentamycin, tobramycin), and colistin. For Gram-positives, the following antibiotic classes were tested (POS32): cefoxitin (screening for MRSA), fluoroquinolones (ciprofloxacin, levofloxacin), aminoglycosides (gentamycin), glycopeptides (teicoplanin, vancomycin), macrolides, lincosamides and streptogramins (erythromycin, clindamycin, clarithromycin), oxazolidinones (linezolid), rifampicin and trimethoprim-sulfamethoxazole.

Antimicrobial susceptibility testing was also performed using the disk diffusion method (Kirby-Bauer, zone diameter), according to the protocol of our hospital laboratory and following European Committee on Antimicrobial Susceptibility Testing (EUCAST) guidelines for interpretation [16]. The tested antimicrobial disks (Oxoid Ltd., Basingstoke, UK) were the following: penicillins (piperacillin-tazobactam 36 μg), cephalosporins (ceftazidime 10 μg, ceftazidime-avibactam 14 μg), carbapenems (imipenem 10 μg), fluoroquinolones (ciprofloxacin 5 μg, levofloxacin 5 μg) and aminoglycosides (amikacin 30 μg, gentamycin 10 μg).

After susceptibility testing for different classes of antibiotics was performed, the microorganisms were identified as MDR if they were resistant to at least one antimicrobial agent in three or more drug classes according to the CDC definition [13]. It must be mentioned that we were able to identify fungi, but our laboratory could not test their susceptibility to antifungals.

### 2.4. Quality Control (QC) and Data Analysis

Quality control for AST by disk diffusion was performed four times per week, according to EUCAST guidelines for internal quality control with *Staphylococcus aureus* ATCC 29213, *E. coli* ATCC 25922, and *Pseudomonas aeruginosa* ATCC 27853 strains [17]. Furthermore, MicroScan QC was performed for the NUC53 panel using *E. coli* ATCC 25922 and *Staphylococcus aureus* ATCC 29213 for the POS32 panel.

Data was stored in Microsoft Office Excel Worksheet, and statistical analysis was performed later in SPSS (bar chart).

### 2.5. Informed Consent Statement

Informed consent was obtained from all subjects involved in this study or their first-degree relatives if the patients had a severe medical status that impaired their ability to reason and give their consent. This study was approved by the Ethics Committee of CEHPRSB (no.3/26.04.2021).

## 3. Results

### 3.1. Patients’ Characteristics

In this study, there were included 202 patients admitted to the ICU of the CEHPRSB from 1 October 2018 to 1 April 2022, from which 758 different bacterial strains were isolated.

Of 202 patients, 122 (60%) were transferred from other hospital units throughout the country. There were 132 (65.35%) male and 70 (34.65%) female patients admitted, with only two patients not having any bacterial infection. The mean TBSA was 35.5%, with a maximum of 95%. The mean TBSA of the survival group is 31% compared to the non-survival group, which is 38.2%. The mean age was 58 years old, with 70% of patients older than 50 years old. Sixty-nine of the 202 patients (34%) survived, with a significant difference between the mean age of the survival group—51 years old compared to the non-survival group—63 years old. Comorbidities were present in 54% (n = 110) of patients. Inhalation injury was present in 45% (n = 91) of the patients. The main patient characteristics are presented in Table 1.

### 3.2. Clinical Specimens and Bacterial Strains

From the collected specimens, there were isolated 737 bacterial strains (Table 2). The most common pathogens in blood were *Pseudomonas aeruginosa* (22 out of 47, 46.81%), followed by *Staphylococcus aureus* (12 out of 47, 25.53%) and *Klebsiella* spp. (5 out of 47, 10.64%). In endotracheal aspirate, most frequently was isolated *Pseudomonas aeruginosa* (84 out of 216, 38.89%), followed by *Candida* spp. (33 out of 216, 15.28%), *Klebsiella* spp. (29 out of 216, 13.43%), *Acinetobacter baumannii* (26 out of 216, 12.04%) and *Staphylococcus aureus* (22 out of 216, 10.19%). The most common pathogens isolated from wound swabs were *Pseudomonas aeruginosa* (165 out of 370, 44.59%), followed by *Staphylococcus aureus* (54 out of 370, 14.59%), *Klebsiella* spp. (39 out of 370, 10.54%), *Acinetobacter baumannii* (32 out of 370, 8.65%) and *Proteus* spp. (24 out of 370, 6.49%). The most common pathogens isolated from urine were *Candida* spp. (41 out of 104, 39.42%), followed by *Pseudomonas aeruginosa* (27 out of 104, 25.96%), *E. coli* (14 out of 104, 13.46%), and *Acinetobacter baumannii* (10 out of 104, 9.62%).

Overall, the most frequent isolated bacterium was *Pseudomonas aeruginosa* (298 out of 737), followed by *Staphylococcus aureus* (88 out of 737), *Klebsiella* spp. (80 out of 737), *Acinetobacter baumannii* (70 out of 737), *E. coli* (35 out of 737), *Proteus* spp. (34 out of 737). Yeasts (*Candida* spp.) were isolated in 92 out of 737 microorganisms (Figure 2).

### 3.3. Antimicrobial Susceptibility Testing Results

There were identified 27 out of 88 *Staphylococcus aureus* strains resistant to oxacillin (31% of *Staphylococcus aureus* strains were MRSA). Of 298 *Pseudomonas* strains isolated, 278 (93%) were MDR. *Acinetobacter baumannii* was the second MDR isolate (n = 64/70), followed by *Klebsiella* spp. (n = 25/80) while *E. coli* had only 2 MDR out of 35 strains (Figure 3).

More than 90% of *Pseudomonas aeruginosa* were MDR, regardless of the clinical specimen from which they were isolated. MDR *Pseudomonas aeruginosa* caused the most bloodstream infections (21/34) among all Gram-negative bacteria. From endotracheal aspirate, MDR *Pseudomonas aeruginosa* was isolated the most (95%), followed by *Acinetobacter baumannii* (85%), other Gram-negative rods (36%), and *Klebsiella* spp. (21%). All *Pseudomonas aeruginosa* and *Acinetobacter baumannii* strains that caused urinary tract infections were MDR (Table 3). Non-fermenter rods that caused wound infections were mostly MDR: *Acinetobacter baumannii* (97%) and *Pseudomonas aeruginosa* (91%). Less than 50% of *Klebsiella* strains were MDR (41% isolated from wounds and 21% from endotracheal aspirates).

The highest resistance rate to colistin was in *Klebsiella* spp., followed at a distance by *Acinetobacter baumannii* and *Pseudomonas aeruginosa*. There was no *E. coli* strain resistant to colistin (Figure 4).

All *Klebsiella* spp. strains isolated from urine were resistant to colistin, and the ones from endotracheal aspirate were 97% resistant (Table 4). *Pseudomonas aeruginosa* strains isolated from endotracheal aspirate had the lowest colistin resistance (4%), followed by those isolated from blood culture (5%); no resistant strain was isolated from urine.

## 4. Discussion

We could underline that identifying the prevalence of bacterial infections and drug resistance in severe burn patients is of very high importance. Two studies performed on burn victims’ autopsies revealed that the main cause of death (up to 65%) is multi-organ failure because of sepsis [18,19]. Sepsis is a common complication in the burn centers ICU, and it is determined by numerous factors, such as length of hospital stay, old age, total burn surface area, severity, and etiology of burns and significantly associated pathologies that lead to immunodeficiencies such as HIV infection or autoimmune diseases.

The main goal of this study, the identification of the pathogenic microorganisms which are causing infections in patients admitted to burn units, was achieved in over 200 patients in a 3.5-year period. *Pseudomonas aeruginosa*, *Staphylococcus aureus*, *Klebsiella* spp., and *Acinetobacter baumannii* are the most frequently isolated bacterial agents in burn patients in our country. These findings are similar to other international studies; however, the proportion of various microorganisms is different between burn centers. In comparison to studies performed in South-East Europe, a Bulgarian study performed almost 30 years ago identified *Staphylococcus aureus* as the main leading agent in burn wound infections [2,3,6,20].

Another goal of this study was to identify the multi-drug resistance profile for the main pathogenic bacteria. We identified a concerning increase in MDR bacteria, especially *Pseudomonas aeruginosa* and *Acinetobacter baumannii*. *Pseudomonas aeruginosa* is mostly susceptible to colistin, also called polymyxin E, which is a 60-year-old antibiotic used as one of the early treatment methods for infections with Gram-negative bacteria, especially *Pseudomonas aeruginosa.* Its use was limited by neuro- and nephrotoxicity and replaced step by step by aminoglycosides [21]. As mentioned in the literature, *Staphylococcus aureus* (MRSA) and *Pseudomonas aeruginosa* are the most found bacteria in burn wounds [2,3,4,5,6,7]. A systematic review made from 481 articles from 2000 to 2017 revealed that glycopeptides antibiotics remain the main weapon in the treatment of MRSA [22]. A retrospective study made between 2007 and 2014 in the burn wards in China to assess the resistance and prevalence of *Pseudomonas* demonstrates increasing concern about the MDR of these bacteria. The percentage of MDR strains of *Pseudomonas aeruginosa* was calculated annually and demonstrated an abrupt curve of resistance, from 64% in 2007 to 89.87% in 2014. The resistance includes amikacin, imipenem, meropenem, cefoperazone-sulbactam, ceftazidime, and ciprofloxacin, and an increasing trend of resistance for the mentioned antibiotics. In some instances, the annual percentage of resistant strains doubled or tripled; for example, only 42.3% of *Pseudomonas* strains were resistant to amikacin in 2007, compared to 85% in 2014 or the resistance to cefoperazone-sulbactam initially 23%, compared to 2014—82% which we find extremely worrying [23].

Because of the emergence of MDR strains, colistin was reintroduced as a line of treatment 10–15 years ago with promising results. Several studies demonstrated its effectiveness on *Klebsiella* spp. and *Acinetobacter baumannii*, as well as on *Pseudomonas aeruginosa* [24]. However, one of the biggest concerns was the dosage of colistin to be used, because there was no consensus in balancing the negative with the positive effects of the drug, the recommendation being to use a maximum of 6 million International Units (I.U.) per day, but other clinicians used 9 million I.U. daily, in patients with normal renal function. Recent studies, on the other hand, demonstrate fewer adverse effects of this antimicrobial agent than older studies [25,26].

Colistin remains our first weapon against Gram-negative bacteria due to its availability in our country and our previous experience with the drug. It is very important to continue to isolate various strains of *Pseudomonas* spp., *Acinetobacter* spp., or *Klebsiella* spp. to identify their susceptibility to antimicrobials, including this drug. In our study, we were lucky to have only 6.71% resistance to colistin in *Pseudomonas aeruginosa* strains which were mostly MDR strains (93%).

A study performed in the Netherlands on a transferred patient with burn wounds from Romania identified six highly resistant Gram-negative bacteria, with five carbapenemase-producing isolates (including *Klebsiella* and *Acinetobacter* strains) and one *Pseudomonas* carbapenem-resistant isolate (Romania being an endemic country for carbapenemase-producing microorganisms) [27]. The isolates were resistant to aminoglycosides, cotrimoxazole, and ciprofloxacin, and one of them was resistant to colistin.

After these findings, we are now undergoing a study on *Pseudomonas aeruginosa* strains to identify acquired resistant genes using phenotypic and genotypic methods, searching for genes that encode the synthesis of extended-spectrum β-lactamase and carbapenemase, such as the study performed in the Netherlands. This practice should be included on a national scale, not only for burn patients, to create a resistance profile of various microorganisms that are responsible for infections in intensive-care units.

We have recently introduced in our practice (2021) ceftazidime-avibactam as an alternative to colistin used, and it is showing great promise. A study performed in 70 United States medical centers on 1900 isolates of *Pseudomonas aeruginosa*, identified ceftazidime-avibactam as the second most susceptible antimicrobial agent against this bacterial infection after colistin (96.9% ceftazidime-avibactam, and 97.5% ceftolozane-tazobactam susceptibility). Piperacillin-tazobactam is the third most susceptible antimicrobial agent for *Pseudomonas,* with 77.5% national effectiveness against these strains [28]. The fact that we are bringing back old treatment methods such as Colistin in our practice means that we are facing a stalemate in our battle with MDR bacterial strains. This means that maybe we should consider preventing, to some degree, the infection by enhancing hospital hygiene worldwide, by designing new methods to cover the wound bed even faster, or by giving preventive medication—the latter has been dismissed in practice by several studies.

Our study is reviving the research in the severe burn field infections and treatment because there has been a lack of data in the last 20 years on this topic in developing countries, such as Romania. Our aim was to inform and draw attention to the importance of acknowledging the problems we are facing in burn wound treatment so that in the future standard care treatment could focus directly on the source of the problem. Unlike most studies that usually investigate pathogen prevalence retrospectively, the strength of this study is its prospective design, being one of the very few made in our country and in South-Eastern Europe, by the number of patients included in the study—202—(this is a big number for severe burn patients in an Intensive Care Unit). Starting from this point, we can address further problems, such as identifying the resistance genes of the strains of *Pseudomonas aeruginosa* and *Staphylococcus aureus*. Additionally, we are undergoing a study on multi-drug resistant bacterial strains from wound isolates by MALDI TOF mass spectrometry as a follow-up from this mothership study.

### Study Limitations

Multi-drug resistance is an ongoing problem in modern societies, and burn treatment is no exception. One of our weaknesses in this study is that we could not perform the above-mentioned research immediately.

The main challenge in the diagnosis of MDR bacterial infection in burn patients is making the distinction between infection and colonization. Although in the past ten years, there has been a blooming scientific field regarding severe burn infections treatment and sepsis diagnosis in this type of patient, there is still no standardization worldwide accepted. We think that a burn center such as ours with over 30 years of experience should contribute with its own opinion and expertise to this scientific field. We acknowledge that our country and our logistic possibilities are less than perfect, and this is also a stalemate in reaching our best potential, for example, the lack of possibilities to collect skin biopsies.

However, in spite of the logistic problems, we have been treating severe burn patients with comparable results to the international data for over 30 years.

Also, the fact that until recently, in our country, we had the only burn-dedicated center nationwide and still have the only severe-burn-dedicated ICU is somehow a minus because we cannot compare our data with other local centers, or to ourselves for this matter, because no studies were performed in the last 20 years.

## 5. Conclusions

In our study, the most frequently isolated microorganisms were *Pseudomonas aeruginosa*, followed by *Staphylococcus aureus*, *Klebsiella* spp., and *Acinetobacter baumannii* in severely burned patients. *Pseudomonas aeruginosa* is our main concern for two reasons: these strains were almost entirely MDR strains, and they were causing the most cases of bloodstream infections, *Pseudomonas* being responsible for over 80% of the fatal outcomes of bloodstream infections.

From the plastic surgeon’s perspective, treating severe burns does not represent the most difficult surgical challenge in our practice, but managing the severely burned patient is certainly the hardest part of our work. Regardless of the good and early surgical management, without having an experienced and homogenous multi-disciplinary team, including infectious diseases specialists, microbiologists, intensive care unit specialists, psychologists, and a nurse-dedicated team, treatment of these patients would be impossible. We must understand the importance of infections and other severe manifestations in burn treatment to unburden our practice and patient management.

This section is mandatory and should contain the main conclusions regarding the research.

## Figures and Tables

**Figure 1 medicina-59-01143-f001:**
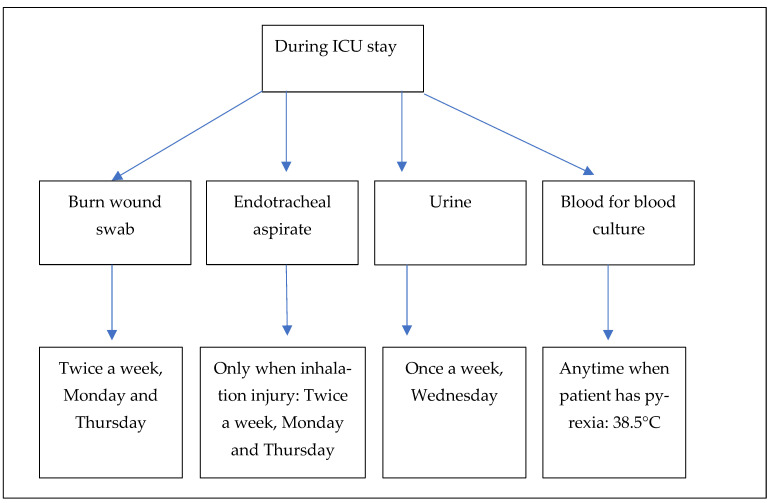
Model of sample collection according to the ICU protocol of CEHPRSB.

**Figure 2 medicina-59-01143-f002:**
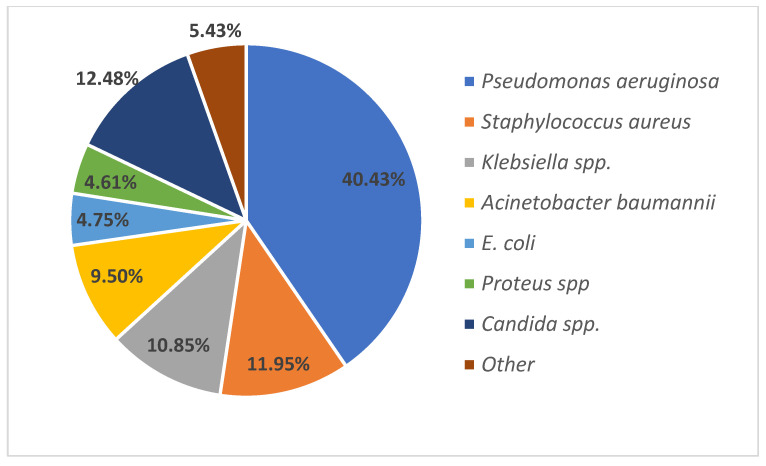
The frequency of isolated microorganisms.

**Figure 3 medicina-59-01143-f003:**
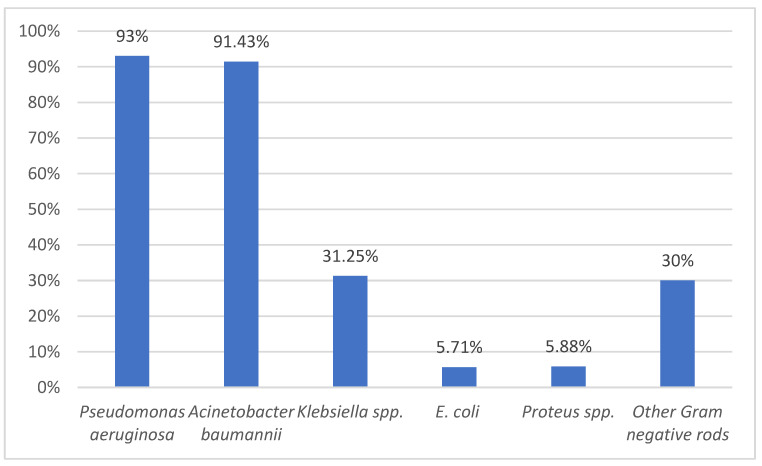
The percentage of MDR strains among each main pathogenic bacterium isolated.

**Figure 4 medicina-59-01143-f004:**
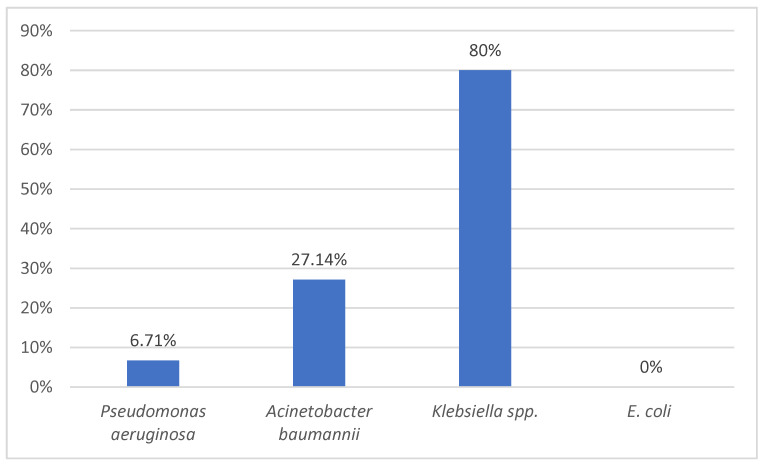
Comparison of colistin resistance among the main Gram-negative rods isolated.

**Table 1 medicina-59-01143-t001:** Patient characteristics—demographics, the severity of the burn, mortality.

Characteristics	Survival (Mean)	Non-Survival (Mean)	Mean (M) or Total (T) for Whole Group
Demographics			
Patients (n)	69	133	202 (T)
Age (years old)	51	63	58 (M)
Sex (male/female)	81/52	51/18	132/70 (T)
Severity			
Comorbidities (yes/no)	77/56	33/36	110/92 (T)
TBSA (%)	31%	38.2%	35.5% (M)
Inhalation injury (yes/no)	26/43	65/68	91/111 (T)

**Table 2 medicina-59-01143-t002:** The isolated microorganisms and their sampling site.

	Wound Swab	Endotracheal Aspirate	Urine	Blood	Total
*Pseudomonas* *aeruginosa*	165	84	27	22	298
*Staphylococcus aureus*	54	22	0	12	88
*Klebsiella* spp.	39	29	7	5	80
*Acinetobacter* *baumannii*	32	26	10	2	70
*E. coli*	18	3	14	0	35
*Proteus* spp.	24	8	1	1	34
Other Gram-negative rods	21	11	4	4	40
*Candida* spp.	17	33	41	1	92
TOTAL bacteria	370	216	104	47	737

**Table 3 medicina-59-01143-t003:** MDR strains and clinical specimens from which they were isolated.

	Wound Swab n/N (%)	Endotracheal Aspirate n/N (%)	Urine n/N (%)	Blood n/N (%)
*Pseudomonas* *aeruginosa*	150/165 (90.91)	80/84 (95.24)	27/27 (100)	21/22 (95.45)
*Acinetobacter* *baumannii*	31/32 (96.88)	22/26 (84.62)	10/10 (100)	1/2 (50)
*Klebsiella* spp.	16/39 (41.03)	6/29 (20.69)	1/7 (14.29)	2/5 (40)
*E. coli*	1/18 (5.56)	0/3 (0)	1/14 (7.14)	-
*Proteus* spp.	1/24 (4.17)	1/8 (12.5)	0/1 (0)	0/1 (0)
Other Gram-negative rods	6/21 (28.57)	4/11 (36.36)	1/4 (25)	1/4 (25)

**Table 4 medicina-59-01143-t004:** Colistin-resistant strains and clinical isolates from which they were isolated.

	Wound Swab n/N (%)	Endotracheal Aspirate n/N (%)	Urine n/N (%)	Blood n/N (%)
*Pseudomonas* *aeruginosa*	16/165 (9.70)	3/84 (3.57)	0/27 (0)	1/22 (4.54)
*Acinetobacter* *baumannii*	5/32 (15.63)	10/26 (38.46)	3/10 (30)	1/2 (50)
*Klebsiella* spp.	25/39 (64.10)	28/29 (96.55)	7/7 (100)	4/5 (80)

## Data Availability

Data are available upon reasonable request to the corresponding authors.

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
