# Peer review of "Etiology and Multi-Drug Resistant Profile of Bacterial Infections in Severe Burn Patients, Romania 2018–2022"

_medicina, 2023, doi:10.3390/medicina59061143_

Round 1

Reviewer 1 Report

Nice descriptive study, but it could be improved.

It would be more interesting to know if the patients were already infected at admission, or if they were infected during their stay in ICU.

Protocol for handling patients infected at admission vs. patients not infected?

Are personel in the ICU routinely screened for carrier status to minimize exposure of vulnerable patients? 

Author Response

Thank you very much for your time and patience. We did our best in addressing almost all of the issues underlined. 

Reviewer 2 Report

Authors focus in a vulnerable population where mortality rates due infectios are high and increasing because multidrug resistance.

Intention to give information about burn infections, which is limited, and scientific community must to pusblish more about it, however, there are several methodological erros which a huge probaility of bias associated since protocol stablished are not the propers according the several guidelines in the international community.

Is very important that work must be revised by microbiological members from the hospital where the research was developed, since some name such as Clostridium are not the currents name (Clostridioides), besides,  as said before, some microbiological procedures as swabs are not within of microbiological recommendations due the difficulty to follow microorganisms with a real association with a infectious process, guideline such publised in 2018 by IDSA (A Guide to Utilization of the Microbiology Laboratory for Diagnosis of Infectious Diseases: 2018 Update by the Infectious Diseases Society of America and the American Society for Microbiology) or manuals prublished and edited by ASM do not recommend swabs, instead the recommendation focus on quantitative biopsies. On the other had, clinical criteria to take blood culture also has associated several bias since guidelines enlisted another clinical parameters to suspect of bacterieamia and sepsis more than temperature, with this parameters, in patients such as burn with inflamatory processes also ca be associated with fever, hypothension, beat rates and all those  enlisted in the guidelines must be revised. When author said urine is recollected, I´d like to know if that urine was taken through a foley catheter?,

Also, microbiological processes deserve a mayor revision since yeast must not been followed in endotracheal aspirate much underreported because this allows that clinicians give treatment to "treat" yeast, in this order participating in selection pressure.

About antibiotics choosen, focus in methodology to check them, authomatized system to determine susceptibility to colistin are no recommended for any guidelines such as CLSI or EUCAST, then, microdilution broth in house must be done for this antibiotic. Any smear evalution was done? Murray and Washington criteria for endotracheal aspirate? For swabs Q Score? How did you decided which microorganisms you must follow and which not?

In table 2, numbers dont fix. For total bacteria in swabs must be 353 and in case you counted Candida must be 370 and total is 387. Pie graphs is not the proper way to show scientific information, may be an histogram could be choosen.

And MDR microorganisms are important but also is important to show in which antibiotics were found the most of resistances.

English is not my native languaje, but in general is ok and understandable

Author Response

(The authors gave the same response as above.)

Reviewer 3 Report

Following are my comments on the text.

- Line 36: TBSA is not Total Burn Surface Area but Total Body Surface Area

- Line 42-45: provide the source publication? And the pathogens of ESKAPE?

- line 62 - is Clostridium difficile, should be Clostridioides difficile

- line 70-74 - Species interactions are indeed a problem. But where? No information about biofilms, because that's where microbes interact with each other.

- line 74-76 - Not really. The whole sentence imprecisely captures the essence of the situation of skin infection. What does it mean: to infect keratinocytes?

- line122-123 - It says "he wounds are dressed in silver 123 sulfadiazine creams." and the correct full name is Sulfonamide silver salts!

- line125 -140 - And why are the blood culture criteria based solely on temperature? Burns often do not present fever, moreover, due to the destruction of tissue masses and massive hypovolemia and disruption of homeostasis, burns present the so-called triad of death: hypothermia, coagulation disorders and acidosis. Where are the criteria for laboratory parameters (CRP, procalcitonin, leukocytosis, etc.)?

General questions about the results and discussion:

1) Is there a correlation between drug resistance profiles and time of sample collection? In patients in whom specific pathogens were isolated, were similar ones isolated from blood/urine/BAL? No statistical study available.

2) How were swabs taken from burn wounds? Biopsy specimens, deep swabs?

3) Colistin is not the only antibiotic of last resort, especially for gram-negative bacilli. Is there data available on other antibiotics?

4) The compilation of the results gives a big dissatisfaction. The results are fragmentary and even out of context.

5) In discussion - in principle, there are many more limits of such studies, it is worth mentioning them. Not only drug resistance is important.

Author Response

(The authors gave the same response as above.)

Reviewer 4 Report

The manuscript encompasses extensive data regarding bacterial infections in burn patients and their resistance profile. The data will impart a great contribution in understanding the reasons of death in burn patients.But there are few aspects which have to be addressed.

The manuscript has to be thoroughly reviewed for English language.

Some sentences are too long (Lines 66-70, 83-87,347-353) to understand

Figure no.4 (which must be figure no 1) seems to be unnecessary.

Figure no.1 (which must be figure no.2) correct 35;5% and 80;11%

Only the resistance against Colistin has been discussed though it is a first line treatment in Romania. It is needed to discuss resistance against other antibiotics briefly.

Author Response

(The authors gave the same response as above.)

Round 2

Reviewer 2 Report

Authors have made the changes cited previously, however, I still have the same problems about the lack about the rigour in microbiological methods according guidelines recommendations.

Candida in endotracheal is not recommended in guidenelines, evidedence is no substantial. Same in urine.

The focus only in hyperthermia allow the lack of several patients with bacteriemia and this is a huge reason of bias.

The use of swabs instead up quantitative biopsies, etc. All this represent a huge cause of bias. Authors say the truth whe the say that, in burned patients, we have a lack of information since we have not made enough research about it, however, the lack of rigourness, unappropiate methodological strategies, do not provides also information.

Author Response

Dear reviewer, 

We did our best to answer to all your comments, hopefully making our paper better. 

Gratefully yours,

The authors of the paper “Etiology and multi-drug resistant profile of bacterial infections in severe burn patients, Romania 2018-2022”

Reviewer 3 Report

Thank you for providing the revised version of the manuscript.

The authors' responses and justifications regarding my comments are satisfactory.

Author Response

Dear reviewer, 

Thank you for your your help in making our paper better. 

Sincerely yours,

The authors of the paper “Etiology and multi-drug resistant profile of bacterial infections in severe burn patients, Romania 2018-2022”
